# Progesterone Receptor Modulates Extraembryonic Mesoderm and Cardiac Progenitor Specification during Mouse Gastrulation

**DOI:** 10.3390/ijms231810307

**Published:** 2022-09-07

**Authors:** Anna Maria Drozd, Luca Mariani, Xiaogang Guo, Victor Goitea, Niels Alvaro Menezes, Elisabetta Ferretti

**Affiliations:** 1Novo Nordisk Foundation Center for Stem Cell Biology (DanStem), University of Copenhagen, 2200 Copenhagen, Denmark; 2Department of Biochemistry and Molecular Biology, University of Southern Denmark, 5230 Odense, Denmark

**Keywords:** progesterone receptor, cardiac differentiation, mesoderm induction, mouse gastrulation, epiblast stem cells, extraembryonic mesoderm, cell adhesion, ECM

## Abstract

Progesterone treatment is commonly employed to promote and support pregnancy. While maternal tissues are the main progesterone targets in humans and mice, its receptor (PGR) is expressed in the murine embryo, questioning its function during embryonic development. Progesterone has been previously associated with murine blastocyst development. Whether it contributes to lineage specification is largely unknown. Gastrulation initiates lineage specification and generation of the progenitors contributing to all organs. Cells passing through the primitive streak (PS) will give rise to the mesoderm and endoderm. Cells emerging posteriorly will form the extraembryonic mesodermal tissues supporting embryonic growth. Cells arising anteriorly will contribute to the embryonic heart in two sets of distinct progenitors, first (FHF) and second heart field (SHF). We found that PGR is expressed in a posterior–anterior gradient in the PS of gastrulating embryos. We established in vitro differentiation systems inducing posterior (extraembryonic) and anterior (cardiac) mesoderm to unravel PGR function. We discovered that PGR specifically modulates extraembryonic and cardiac mesoderm. Overexpression experiments revealed that PGR safeguards cardiac differentiation, blocking premature SHF progenitor specification and sustaining the FHF progenitor pool. This role of PGR in heart development indicates that progesterone administration should be closely monitored in potential early-pregnancy patients undergoing infertility treatment.

## 1. Introduction

One of the leading causes of recurring early pregnancy loss is embryo implantation failure and consequent embryonic death [1]. Hormonal treatment such as progesterone is frequently used to stimulate and support pregnancy, with maternal tissues being its primary target [2]. Progesterone signaling is evolutionarily conserved in mammals, and progesterone receptor (PGR) was shown to support functions of the adult organs such as the brain and heart [3,4,5]. While addressing its function in human embryo is impossible for ethical reasons, in mice, PGR is expressed in the gastrulating embryo, raising questions about its function during embryonic development. 

Furthermore, progesterone is also associated with early murine embryonic development, as low progesterone concentrations lead to the developmental delay of blastocysts in vitro [6]. Following implantation, gastrulation represents a key step in embryonic life. In mammals, gastrulation is marked by a dynamic and transient population of cells located in a region called the primitive streak (PS), which gives rise to the mesoderm and the definitive endoderm [7,8,9]. Cells migrating out of the posterior end of the PS, corresponding to the posterior proximal end of the embryo, form the extraembryonic mesoderm that supports embryonic growth [7,9]. Extraembryonic mesoderm will give rise to the allantois, amnion, primitive hematopoietic cells, and the yolk sac. Conversely, cells emerging from the anterior PS will generate the embryonic mesoderm and definitive endoderm. The mesoderm will contribute to essential organs such as the heart, facial and skeletal muscles, bones, gonads, and kidneys, while the endoderm forms the pancreas, lungs, liver, and gut [8,9].

Whether progesterone could have a dual role in supporting maternal tissues and embryonic development at gastrulation is unknown. We found that PGR is expressed in a posterior–anterior gradient in the PS of gastrulating murine embryos, resembling the BMP signaling gradient. We established in vitro cell culture systems to model gastrulation and assessed the role of PGR in post-implantation cells. We employed mouse epiblast stem cells (EpiSCs) that correspond to the undifferentiated epiblast of post-implantation embryos and are primed to differentiate in vitro into different germ layer lineages [10,11]. To simulate the spatial allocation, we used various combinations of signaling cues that recapitulate the environment present in the embryo at gastrulation, mimicking the anterior (APS) and posterior (PPS) regions of the PS [12], and employed scRNA-Seq analyses to assess differentiation efficiency. 

We found that PGR is expressed in EpiSCs and demonstrated that it is involved in early mesoderm specification, modulating extraembryonic and cardiac mesoderm differentiation. Furthermore, we show that PGR has a specific effect on the maturation of the PS-derived progenitors, promoting the differentiation of the BMP-responding cell populations and thus favoring the posterior mesoderm progenitors such as extraembryonic mesoderm and first heart field. We also report that during differentiation, progesterone signaling differentially modulates the expression of actin cytoskeleton remodeling, focal adhesion, and cell motility proteins implicated in the process of mesoderm formation. 

In conclusion, we demonstrate that PGR plays a role in the early specification of extraembryonic mesoderm and cardiac progenitors at gastrulation. Given progesterone is used as a treatment for recurring early miscarriages and improved in vitro fertilization (IVF) success rate, the offspring of patients undergoing infertility treatment should be monitored with special emphasis on cardiovascular conditions.

## 2. Results

### 2.1. The Nuclear Receptor PGR Is Expressed in Primed Pluripotent Cells

Primed pluripotency is the state that immediately precedes germ layer specification, including PS induction and mesoderm differentiation [13,14]. We speculate that transcription factors (TFs) supporting the primed state could be important for setting up epiblast for early lineage commitment, biasing the choice towards specific cell lineages and cell types, such as the mesoderm progenitors. We also hypothesized that these TFs should be expressed in the early stage of PS induction and will play a subsequent role in early germ layer specification. The plasticity of epiblast cells ingressing at the PS is progressively restricted by BMP/TGF-beta/WNT signaling (Figure 1A,B) [14,15]. To simulate the spatial allocation, we use different combinations of signaling cues to recapitulate the environment present in the embryos at gastrulation time, mimicking the anterior (APS) and posterior (PPS) regions of the PS (Figure 1C). RNA-Seq profiling shows that PPS cells are enriched in posterior streak markers, *Evx1*, *Bmp4*, *Hand1*, *Tbx3*, *Foxf1*, and *Pgr*, demonstrating that a high level of BMP restricts the fate of PPS towards posterior and extraembryonic mesoderm, while the anterior markers *Gsc*, *Sox17*, *Mesp1*, *Tbx6*, *Eomes*, and *Foxa2* are more abundant in the APS cells, suggesting that this population is enriched in progenitor cells with endoderm, cardiac, and paraxial mesoderm signatures.

To uncover TFs influencing the differentiation potential of primed pluripotent cells, we compare the transcriptional profiles of EpiSCs and mESCs cultured in either 2i/Lif or serum/Lif and identify 19 TFs enriched in EpiSCs (Figure 1D) [14]. After evaluating the known knockout phenotypes and functions of the short-listed genes, we narrowed the list down to six candidates *(Bcl11b*, *Pgr*, *Tshz1*, *Zfp521*, *Prox1*, and *Zfhx4*) (Figure 1E and Appendix A). The expression analysis during PS induction and differentiation towards mesoderm-specific lineages (extraembryonic, lateral, and paraxial mesoderm) highlighted that two TFs are upregulated in PS: *Pgr* and *Bcl11b* (Figure 1E), suggesting their involvement in priming EpiSCs towards mesodermal or endodermal cell fate. Interestingly, progesterone receptor encoded by the *Pgr* gene is historically studied in the context of breast cancer, endometriosis, and embryo implantation [16,17,18,19,20]. However, our in vitro expression data suggests a novel function in the post-implantation epiblast during mesoderm specification. By comparing the expression pattern observed in vitro with the published scRNA-Seq data of gastrulating mouse embryos [21], we found that *Pgr* is expressed in the epiblast, PS, and mesoderm in vivo (Figure 1F). These data resemble the expression patterns we observe with the in vitro cell culture systems. 

**Figure 1 ijms-23-10307-f001:**
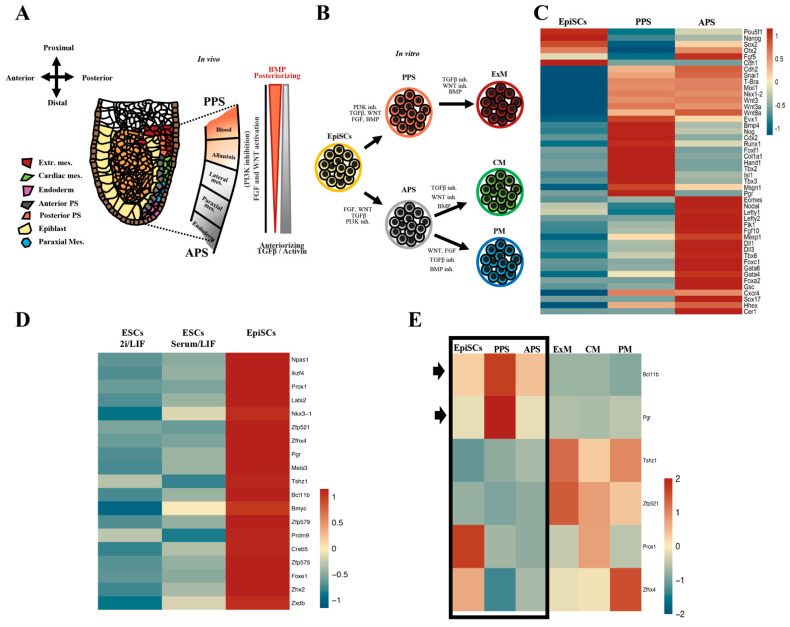
Identification of PGR as a TF candidate involved in priming EpiSCs towards mesoderm fate. (**A**) Schematic representation of E6.5–7.0 gastrulating mouse embryo. Epiblast cells (yellow) ingress and become the posterior (PPS, light red) and/or anterior (APS, grey) primitive streak (PS). Cells originating in different regions of the PS display distinct mesodermal fate bias. Extraembryonic mesoderm (dark red) including blood (hemangioblast) and allantois originate from PPS, while APS will give rise to cardiac mesoderm (green), paraxial mesoderm (blue), and definitive endoderm (pink). The coordinated activity of different signaling pathways (NODAL, WNT, and BMP) activate specific transcriptional programs that direct mesodermal fate. (**B**) Illustration summarizing the protocol used to differentiate EpiSCs towards PPS and APS cells and their derivatives, cardiac mesoderm (CM) and paraxial mesoderm (PM) progenitors. (**C**) Heatmap of RNA-Seq of EpiSCs, PPS, and APS delineating the antero-posterior identity of the in vitro-generated cell populations. PPS cells are enriched in extraembryonic markers and known BMP target genes (*Evx1*, *Bmp4*, *Hand1*, *Foxf1*, and *Tbx3*), while APS cells show high levels of anterior PS markers (*Gsc*, *Sox17*, *Tbx6*, *Eomes*, and *Foxa2*). (**D**) Identification of 19 TFs specifically enriched in EpiSCs by comparing RNA-Seq data of EpiSCs with mouse ESCs cultured in 2i/LIF and serum/LIF. (**E**) Heatmap showing the expression of the six short-listed TFs in different cell types. *Blc11b* and *Pgr* exhibit a distinct pattern of expression, with the highest levels of mRNA found in EpiSCs, PPS, and APS (highlighted by arrows), (RNA-Seq, *p* ≤ 0.05, fold-change ≥ 1.5). ESC, embryonic stem cells; EpiSC, epiblast stem cells; PPS, posterior primitive streak; APS, anterior primitive streak; ExM, extraembryonic mesoderm; CM, cardiac mesoderm; PM, paraxial mesoderm. (**F**) UMAP visualization and clustering of scRNA-Seq of gastrulating mouse embryos showing *Pgr* expression in the epiblast (yellow circle) and mesoderm (brown circle) (data set obtained from Pijuan-Sala et al. Nature 2019 [21]).

### 2.2. PGR Is Expressed in the Primitive Streak with a Posterior-to-Anterior Gradient

To further characterize *Pgr* function during gastrulation, we assess PGR protein levels and localization in embryonic day (E) 6.0–6.5 mouse embryos. PS formation represents the initiation of gastrulation, the process in which ectoderm, mesoderm, and endoderm cell fates are established [7,8]. The time and location of cells emerging from the PS are linked with subsequent cell identity, with extraembryonic mesoderm originating from the PPS, while cardiac mesoderm, paraxial mesoderm, and definitive endoderm cells arise from the APS (Figure 2A). Immunofluorescence (IF) shows that PGR is present at high levels in the PPS (Figure 2B, panels I–II), with a lower amount in APS (Figure 2B, panel II), and strikingly, absent in the anterior region of the gastrulating E6.75 embryo (Figure 2B, panel III). The embryo is co-stained with antibodies against SOX2 and T-BRA to identify the epiblast and the PS, respectively [22]. These data support the hypothesis that PGR plays a role in PS induction and potentially in mesoderm formation. Next, we employ mouse EpiSCs as cell culture model and analyze PGR levels during in vitro differentiation. Western blotting reveals higher levels of PGR in the PPS compared with the EpiSCs and mesoderm progenitors, extraembryonic mesoderm, and cardiac and paraxial mesoderm (Figure 2C), suggesting a correlation between PGR protein expression in vitro and in vivo. 

PGR is a nuclear receptor that translocates from the cytoplasm to the nucleus upon ligand binding and acts as a TF to either activate or repress the expression of its target genes [23,24]. However, ligand-bound PGR also has cytoplasmatic functions. Therefore, we use IF to pinpoint the cellular localization of PGR in EpiSCs and PPS. As shown in Figure 2E, the levels of PGR are higher in PPS, with a strong signal indicating cytoplasmatic localization. However, a small proportion of the receptor is found in the cell nucleus, indicating that it could also exert its function directly as a TF. By comparing *Pgr* expression among different lineages, we confirm that *Pgr* is highly expressed in PPS compared with definitive endoderm and neural lineages (Figure 2F). Significantly, RT-qPCR reveals that *Pgr* expression peaks at 18–24 h after inducing PS, which corresponds to a high expression of *T-Bra*, the hallmark PS marker (Figure 2F) [22,25]. We therefore show that the expression pattern of PGR in vitro corresponds to the one we report in vivo. 

**Figure 2 ijms-23-10307-f002:**
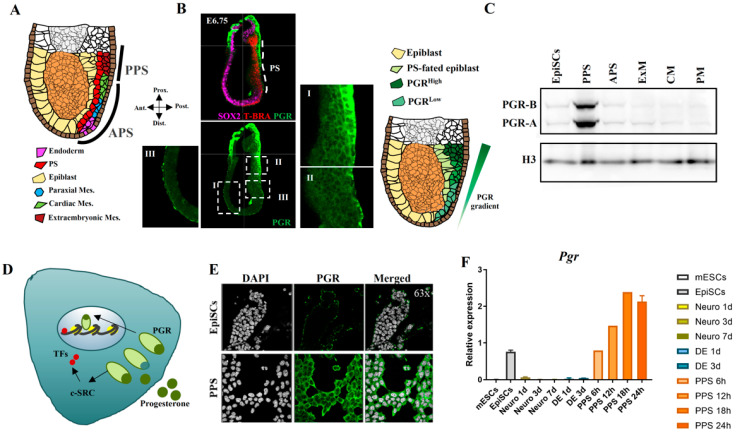
*Pgr* is expressed in primitive streak cells in vivo and in vitro. (**A**) Illustration depicting E6.5-E7.0 gastrulating embryo and indicating posterior primitive streak (PPS), anterior primitive streak (APS), and their derivatives. Endoderm (pink), primitive streak (red), epiblast (yellow), paraxial mesoderm (blue), cardiac mesoderm (green), and extraembryonic mesoderm (dark red). (**B**) Confocal images of E6.75 embryos probed with T-BRA (red), SOX2 (magenta) and PGR (green) antibodies reveal a gradient of PGR with higher levels in PPS. Panel I shows high levels of PGR in the PPS, panel II demonstrates low levels of PGR in the APS, while in panel III the complete absence of PGR in the most anterior part of the embryo. The schematic on the right depicts the PGR gradient (green) in the posterior and anterior PS. Images were taken using 40× objective. (**C**) Western blot assay confirming that PGR protein levels follow the same pattern as mRNA during mesoderm induction, with the highest protein concentration in PPS cells compared with EpiSCs, APS, ExM, CM, and PM. Histone H3 (H3) is used as loading control. (**D**) Illustration depicting the dual PGR activity as a ligand-dependent TF (nuclear localization) and modulator of signaling pathways (cytoplasmatic localization) with the previously described interaction with c-SRC [24]. (**E**) Immunofluorescence showing PGR localization in EpiSCs and PPS. In the EpiSCs, low levels of PGR are detected. In PPS, PGR protein is localized mainly in the cytoplasm with a smaller percentage present in the nucleus. Images were taken using 63× objective. (**F**) RT-qPCR analysis of *Pgr* expression across different cell lineages. *Pgr* transcript is specifically expressed in EpiSCs and PPS. Time course induction of PPS shows the highest expression at 18 h of differentiation.

### 2.3. PGR Modulates the Differentiation of the Posterior Primitive Streak and Extraembryonic Mesoderm 

Considering the unique expression pattern of *Pgr* in vivo and in vitro, we address its role in the specification of the PPS and its derivative, the extraembryonic mesoderm [26]. We first established a protocol generating PPS starting from EpiSCs and subsequently differentiated PPS towards the extraembryonic mesoderm (Figure 3A). Using bulk RNA-Seq, we profiled the extraembryonic mesoderm. These data confirm that the differentiation protocol recapitulates the expression pattern observed in the mouse embryo (Figure 3B). A further characterization by scRNA-Seq reveals that the PPS is a homogeneous cell population expressing the posterior markers *Evx1*, *Hand1*, *Foxf1*, and *Col1A1* (Figure 3C and Appendix A). Similarly, scRNA-Seq analysis of extraembryonic mesoderm reveals a homogeneous distribution of the ExM markers *Hand1*, *Igfr2*, *Bmp4*, and *Foxf1* [27,28] (Figure 3D and Appendix A). These data demonstrate the efficiency of the extraembryonic mesoderm differentiation protocol. We next generate EpiSC lines lacking *Pgr* via CRISPR-mediated knock-in of a hygromycin resistance cassette. This approach produces a truncated *Pgr* mRNA that is subsequently degraded (Appendix A). The *Pgr* Knockout clones (*Pgr-KO*) are screened by PCR genotyping, and the lack of the receptor protein is confirmed by Western blot (Appendix A). We identified at least three clones lacking *Pgr* expression with one allele with successful integration of the hygromycin cassette and one with CRISPR-induced frameshift that generates multiple premature stop codons (Appendix A). To confirm this, similarly to the WT cells, these mutated clones maintain their pluripotent and post-implantation epiblast characteristics, we assessed the expression of the pluripotency and epiblast markers such as *Oct4*, *Sox2*, *Nanog*, and *Otx2* [29] (Appendix A). 

Next, we differentiate WT and *Pgr-KO* EpiSCs towards PPS and extraembryonic mesoderm. In PPS, the extraembryonic signatures, connected to BMP signaling, *Bmp2*, *Bmp4*, and *Hand1*, are strongly affected, while other PPS markers, *T-Bra*, *Mixl1*, and *Evx1*, are slightly influenced (Figure 3E). The ExM markers *Foxf1*, *Bmp4*, and *Hand1* [27,28,30,31], are strongly downregulated due to *Pgr* inactivation (Figure 3E), suggesting that extraembryonic mesoderm formation is affected in *Pgr-KO*. Conversely, the upregulation of *Tal1* and *Gata2* could indicate a shift of *Pgr-KO* extraembryonic mesoderm cells towards hemangioblast fate [31,32]. *Foxf1* and *Pitx1* are markers of allantois progenitors in the gastrulating embryos [21].They are downregulated in the mutant cells, supporting the role of PGR in extraembryonic mesoderm differentiation and its function in supporting allantois over hemangioblast specification. In conclusion, we demonstrate that while *Pgr* is dispensable for maintaining the EpiSCs state, it is important for PPS induction [33], and it ultimately plays a critical role in the formation of extraembryonic mesoderm [34].

### 2.4. PGR Influences the Acquisition of the First Heart Field Fate

Many genes involved in extraembryonic mesoderm differentiation, such as *Hand1*, *Foxf1*, and *Bmp4*, are also important for cardiac differentiation [28,35]. These genes appear to be regulated in a *Pgr*-dependent manner during extraembryonic mesoderm differentiation, suggesting a similar role for *Pgr* in regulating the aforementioned genes during cardiac differentiation. Thus, to assess if the lack of PGR affects cardiac differentiation, we establish a differentiation protocol using EpiSCs, initially inducing APS followed by cardiac mesoderm formation (Figure 4A,B). Two progenitor cell types, termed first heart field (FHF) and second heart field (SHF), contribute to the developing heart. The earliest cardiac progenitors that are specified form the FHF and build the left ventricle with a small contribution to the atria. SHF cells arise later and form the right ventricle, atria, and outflow tract [36]. These progenitors are produced in vitro with our differentiation protocol [36,37] (Figure 4B). RNA-seq analysis of WT and *Pgr-KO* cells confirms that extraembryonic and cardiac mesoderm generated from the mutant cells significantly differs from the WT counterparts (Figure 4C,D). Interestingly, the differentiation of *Pgr-KO* cells towards cardiac mesoderm clearly favors the SHF-fated cells with cardio-pharyngeal signature identified by the upregulation of *Gata4*, *Nkx2-5*, *Hand2*, *Mef2c*, and *Tbx1* markers [37] (Figure 4E). In contrast, downregulation of *Hand1*, *Foxf1*, and *Flk1* markers indicates that *Pgr-KO* cells fail to produce the FHF cardiac progenitors. Of note, the reduction in *Hcn4* and *Tbx3*, which label the heart conductive system progenitors and are FHF derivatives, reiterates PGR’s role in FHF specification [38,39,40]. At the same time, late cardiomyocyte markers such as *Myl4*, *Myl7*, *Tnnt2*, and *Tnni1* are upregulated in *Pgr-KO* cells (Figure 4E), suggesting that PGR plays a role in the maturation of cardiac muscle cells. Taken together, these data indicate that in addition to its role in extraembryonic mesoderm differentiation, PGR is involved in promoting FHF cell fate and inhibiting SHF progenitor specification. 

### 2.5. PGR Modulates Mesoderm Differentiation in an Isoform-Dependent Manner

To further investigate the role of PGR in the specification of the APS and cardiac progenitors, we performed overexpression experiments. PGR exerts its function via two distinct isoforms generated using alternative promoters [41]. PGR-B is a full-length protein, while PGR-A is shorter, lacking the N-terminal transactivation domain [42] (Figure 5A). In addition to common molecular activities in the uterine-derived epithelial cells [1], PGR isoforms were shown to have distinct functions during embryo implantation. To understand which of the PGR isoforms is involved in cardiac specification, we prepared lentiviral overexpression (OE) constructs (Figure 5B). In all three cases (PGR-B OE, PGR-A OE, and PGR-B+A OE), *Pgr* isoform mRNA levels are at least four-fold higher than control EpiSCs (Figure 5C). Control primers targeting 3′UTR were included to indicate endogenous transcript level. The efficient translation of the OE constructs is confirmed by Western blot (Figure 5D). All three cell lines overexpress the correct protein isoform at higher levels than EpiSCs, APS, and PPS, where PGR has the highest expression levels. The differentiation of PGR-B OE and control EpiSCs towards cardiac mesoderm shows only a mild upregulation of the FHF markers *Hand1*, *Flk1*, and *Foxf1*, indicating that PGR-B could directly or indirectly modulate FHF progenitors (Figure 5E). In contrast, the strongly decreased expression of SHF markers *Nkx2-5*, *Hand2*, and *Gata4* in cells overexpressing PGR-A indicates that PGR inhibits SHF fate solely or mainly through its short isoform (Figure 5F). In conclusion, we show that PGR modulates cardiac mesoderm differentiation in an isoform-dependent manner. 

### 2.6. PGR Influences Cell–ECM Interaction and Cytoskeletal Remodeling

Extracellular matrix (ECM) binding, cell adhesion, cytoskeleton remodeling, and cell motility are mechanistic factors strongly influencing PS induction and mesoderm formation [43]. PGR activity was implicated in increasing cell motility and cytoskeletal remodeling [44], suggesting that PGR could control these processes during gastrulation. Indeed, Gene Ontology (GO) enrichment analysis on WT vs. *Pgr-KO* in PPS cells highlights terms associated with cell adhesion and motility (Figure 6A). RNA-seq data show the altered expression of specific integrins, collagens, and actin-binding proteins, especially in PPS and APS. These proteins are involved in cell migration during PS formation. Interestingly, PS markers such as *T-Bra* and *Mixl1* are not strongly impacted, suggesting that changes in adhesion and motility do not alter PS identity in *Pgr-KO* cells. When PS cells are specified into mesoderm progenitors, changes in ECM–cell and cell–cell interactions are essential to generate the physical environment and chemical gradient driving signaling response [45]. RNAseq analysis shows that ECM proteins such as Laminins and Collagens are upregulated in extraembryonic and cardiac mesoderm (Figure 6B), while integrin expression indicates a cell-type-specific pattern, with *Itga3* and *Itga4* differently expressed in extraembryonic mesoderm and cardiac mesoderm, respectively. A comparison of WT and *Pgr-KO* transcriptional profiles shows changes in laminins and cell-specific integrin expression in the extraembryonic and cardiac mesoderm (Figure 6B). While the majority of integrins are downregulated in *Pgr-KO*, *Itgb3* is specifically upregulated in the extraembryonic mesoderm. 

Furthermore, in cardiac mesoderm, Ncam1 [46], the cadherin involved in the cell–cell interactions of atrial and ventricular cells is upregulated. These data suggest that PGR could influence cell differentiation directly or indirectly by selectively modulating the specific mechanisms underlying cell–ECM and cell–cell interactions as well as cytoskeleton remodeling (Figure 6C). 

## 3. Discussion

The importance of progesterone signaling in preparing and supporting pregnancy as well as maintaining ovulation is well-documented [2,18,20,47,48]. Recently, it was also reported that PGR supports in vitro development of murine pre-implantation blastocysts [6]. 

We reported that, in vivo, PGR is expressed in the PS of gastrulating mouse embryos in a posterior-to-anterior gradient, and enriched in primed pluripotent stem cells, EpiSCs, which represent the cellular state preceding gastrulation. In vitro, during EpiSCs differentiation towards mesoderm, both the transcript and protein peak in a 24 h window, which coincides with the high expression of the PS marker *T-Bra*, supporting a potential role in mesoderm differentiation.

Here, we report that PGR is involved in regulating extraembryonic and cardiac mesoderm differentiation. We show that PGR has a lineage-specific effect on the maturation of extraembryonic mesodermal progenitors, promoting hemangioblast differentiation over allantois fate.

Additionally, during cardiac differentiation, PGR promotes the expression of the FHF markers *Hand1* and *Foxf1*, as well as *Hcn4* and *Tbx3*, which label the heart conductive progenitors, suggesting that it could support the formation of atrial and ventricular cardiomyocytes with FHF origin. Interestingly, PGR has an opposite effect on SHF cardiac genes, inhibiting the expression of *Hand2*, *Mef2c*, *Myl4*, *Myl7*, *Tnnt2*, and *Tnni1*. These results suggest that PGR promotes the differentiation of FHF derivatives and represses SHF fate. While *Pgr* mutant embryos do not have evident cardiac defects, it is possible that an alternative hormonal receptor compensatory mechanism exists in vivo. Another possibility is that SHF-derived cells could rescue the FHF deficiency. In zebrafish embryos, ablation experiments showed that SHF could compensate for the absence of FHF [49].

The observation that the in vitro phenotype is more severe in *Pgr-KO* cells exposed to differentiation conditions towards extraembryonic and cardiac mesoderm that rely on BMP signaling suggests that PGR modulates BMP-driven mesoderm specification. The posterior–anterior gradient of PGR in the embryos also resembles BMP levels at gastrulation, supporting a possible BMP–PGR interaction. It has previously been reported that BMP signaling upregulates PGR expression in the uterus. Similarly, BMP could induce PGR in the posterior PS [50]. Therefore, specification of FHF progenitors could be mediated by a BMP–PGR driven mechanism, with SHF arising from cells exposed to a lower level of BMP and PGR. Strikingly, it has been shown previously that high BMP activity is required to induce *Gata4* and *Nkx2-5* expression at E8.0, but not for their maintenance, and that the BMP receptor is crucial for the specification of the FHF. SHF progenitor differentiation instead strongly relies on canonical WNT signaling [51]. Moreover, it was reported that with in vitro differentiation models, BMP signaling directly activates the transcription of *Id1*, which is critical for the cardiac induction of FHF genes [52]. On the contrary, SHF genes are not subject to ID1-BMP-mediated regulation, supporting the hypothesis that FHF and SHF have different requirement for BMP and PGR. In another context, progesterone and BMP have previously been associated as inducers of branching morphogenesis in epithelial mammary cells [53]. 

The dual role of PGR as cardiac progenitor modulator is clearer in the overexpression experiments. While overexpression of PGR has a moderate effect on FHF markers, it has, on the contrary, a strong effect on SHF, suggesting that PGR blocks SHF progenitors if present at higher levels. We can speculate the existence of a BMP-dependent safe-lock mechanism blocking premature SHF differentiation or transdifferentiation and sustaining concomitantly the FHF progenitor pool. To work as a safeguard of premature differentiation could be the major role of PGR, and it could explain why the *Pgr* mutant mice do not have strong cardiac defects. 

FHF cells mostly become cardiomyocytes, while SHF-derived cells are multipotent and give rise to various cardiac cell types, such as cardiomyocytes, smooth muscle, endothelial, and fibroblast cells. In adult mice, it was reported that progesterone has an anti-atherosclerotic effect and inhibits proliferation and migration of aortic smooth muscle cells [54,55]. Thus, PGR could have an inhibitory effect on cardiac muscle cells originating from the SHF^57^. Again postnatally, progesterone supplementation promotes neonatal cardiomyocytes proliferation through the upregulation of YAP signaling [56]. Thus, PGR could promote cardiomyocyte formation.

Mesoderm delamination from the epiblast requires basal membrane disruption, apical constriction, loss of apicobasal polarity, changes in cytoskeletal properties, and acquisition of cell motility. Thus, after exiting pluripotent epiblast state, the different mesodermal populations acquire a specific set of cytoskeletal components necessary for cell motility and adhesion, which can underlie the differences in their migratory properties. A recent study showed that during mouse gastrulation, extraembryonic and embryonic mesoderm showed differences in cell migratory properties [57]. ExM and embryonic mesodermal cells have distinct shapes, cytoskeletal composition, and migration dynamics that might play a role in determining their ultimate fate [58]. We found that integrins *Itga3* and *Itga4* are specifically expressed in extraembryonic and cardiac mesoderm, respectively, indicating that specific cell–matrix interactions could differentially affect their migratory and differentiation properties. Integrins can signal across the plasma membrane in both directions, transducing extracellular and intracellular signals affecting cell fate [58]. ITGB3 mediates the cross-talk between cells and the stromal microenvironment in different contexts by direct interaction with fibronectin and focal adhesion proteins but also by vesicle-based intercellular communication [59]. In *Pgr* mutant cells, the upregulation of *Itgb3* in ExM suggest that a PGR-ITGB3-dependent mechanism could mediate the establishment of a specific microenvironment promoting extraembryonic mesoderm differentiation. ITGB3 is historically associated with hemangioblast differentiation, and its upregulation coincides with increased levels of *Tal1* and *Gata2*, thereby indicating a shift of *Pgr-KO* ExM cells towards a hemangioblast fate [60]. These data suggest that acquisition of the mesodermal state correlates with ECM interaction and the establishment of a microenvironment that mediates specific cell responses and differentiation.

Altogether, the observed phenotypes could be attributed to the previously described PGR–cSRC interaction [24]. The perturbation within the PGR–cSRC network could alter the downstream PI3K/AKT and MAPK signaling pathways affecting proliferation and differentiation, or the JNK, RhoA, and FAK cascade influencing cell motility and adhesion (Figure 6C). As a consequence, changes in adhesion properties and ECM composition could lead to the reshaping of the cellular niche, thereby indirectly influencing cell fate (Figure 6C). Therefore, PGR could act as a BMP-induced modulator of cell fate through c-SRC, fine-tuning and amplifying the response to FGF and thereby generating a gradient of response (Figure 7A,B).

In other contexts, PGR is known to directly affect cytoskeleton actin dynamics and focal adhesion formation in breast cancer, ovarian, neural, and human endothelial cells. Furthermore, previous reports showed that progesterone is involved in cytoskeletal remodeling and cell migration during gastrulation in zebrafish embryos [61]. While this study focused mainly on the ligand and enzymes involved in its synthesis, we demonstrated that during gastrulation, progesterone signaling exerts its function primarily through its receptor. PGR indeed affects the expression of genes related to cell adhesion and microtubule dynamics. 

The PGR protein exists in at least two isoforms—a longer PGR-B and an N-terminally truncated PGR-A [41,42]. The balance of the PGR isoforms is important to promote the transition into partuition. PGR-A overexpression induces smooth muscle contractility of the uterus [62]. Similarly, during cardiac mesoderm specification, different PGR isoform activity could indicate a specific requirement in contractility of FHF and SHF cardiomyocytes. Notably, we show that while PGR-B mildly modulates the expression of FHF markers, PGR-A has a major effect inhibiting SHF genes. We also report that its absence promotes differentiation towards hemangioblast and SHF fates, and that these functions are carried out in an isoform-dependent manner. These data indicate that the alternative isoforms exhibit overlapping and distinct functions in the different mesodermal progenitors.

Progesterone and its synthetic analogues are used as contraceptive agents [62] and a treatment for recurring early miscarriages [63]. Our work in mouse and cell models shows that PGR is essential not only for embryo implantation but also for embryonic development, thus further emphasizing its significance during pregnancy supporting embryonic life. Moreover, our work suggests monitoring progesterone administration during early pregnancy so as not to impact embryonic heart development negatively.

The potential impact of this study, however, extends beyond the consequences for embryonic development. Stem-cell-based therapies are considered to be one of the possible groundbreaking solutions for degenerative diseases such as type 1 diabetes, Parkinson’s, and heart failure, as they can be differentiated into virtually any somatic cell type [64,65,66]. Although great progress is being made in this field, differentiation protocols require further improvements to produce uniform populations of desired cells with high efficiencies. Our study sheds some light on the mechanisms involved in mesoderm development, focusing specifically on cardiac fate commitment and shows that the level of progesterone in the cell culture medium could be one of the factors influencing in vitro differentiation.

## 4. Materials and Methods

Unless stated otherwise, all chemical reagents were purchased from Sigma Aldrich, all cell culture reagents from Thermo Fisher Scientific (Waltham, MA, USA), and all the cytokines from Qkine (Milton, Cambridge, UK).

### 4.1. Cell Culture

A total of 129 EpiSCs were cultured on fibronectin-coated plates (15 µg/mL; Merck Millipore (Burlington, MA, USA), FC01015) in N2B27 supplemented with 20 ng/mL Activin A Plus, 10 ng/mL bFGF and 1 µM XAV939 (EpiSC medium). N2B27 was prepared in the following way: 200 mL DMEM:F12, 200 mL Neurobasal Medium, 4 mL L-Glutamine, 400 µL β-mercaptoethanol, 2 mL N2 supplement, and 4 mL B27 supplement. N2 reagent was prepared by adding 10 mL 7.5% BSA to 68.75 mL DMEM:F12, mixing well, and adding the following components drop-wise: 250 mg insulin re-suspended in 10 mL 0.01 HCl, 1 g apo-transferrin dissolved in 10 mL MilliQ water, 330 μL progesterone (stock solution 0.6 mg/mL), 1 mL putrescine (stock solution 160 ng/mL), and 100 μL Na selenite (stock solution 3 mM). Mouse embryonic stem cells (mESCs) were cultured on gelatin-coated plates in N2B27 medium supplemented with LIF, 1 μM PD-0325901, and 3 µM CHIR99021. As part of standard cell culture procedure, all cell lines were mycoplasma-tested and confirmed to have normal karyotype.

### 4.2. EpiSC Differentiation to PS and Mesoderm Progenitors

Cells were differentiated in a monolayer and seeded on fibronectin-coated wells at 10.5 × 10^3^ cells/cm^2^ in chemically defined EpiSC medium 24 h prior to induction of differentiation. PPS cells were induced by supplementing N2B27 with 30 ng/mL Activin A, 40 ng/mL BMP4, 6 µM CHIR-99021, 20 ng/mL bFGF, and 100 nM PIK-90, while APS cells induction medium consisted of N2B27 with 30 ng/mL Activin A, 4 µM CHIR-99021, 20 ng/mL bFGF, and 100 nM PIK-90. In both cases, cells were cultured in the appropriate media for 24 h. PPS cells were further differentiated to extraembryonic mesoderm (ExM) for another 24 h by supplementing N2B27 with 1 μM A-83-01, 30 ng/mL BMP4, and 1 μM C59. Lateral mesoderm (LM) was obtained by applying the ExM medium on APS cells for 24 h, and finally paraxial mesoderm (PM) was induced from APS with N2B27 supplemented with 1 µM A-83-01, 3 µM CHIR-99021, 250 nM LDN193189, and 20 ng/mL bFGF for 24 h.

### 4.3. RT-qPCR

Cells were washed once with PBS and detached from the culture dish using Accutase. The cells were collected, and RNA was extracted using QIAGEN RNeasy Mini Kit (Qiagen, 74106, Hilden, Germany) according to the manufacturer’s instructions. A total of 1 μg of RNA was reverse transcribed using SuperScript III enzyme (Thermo Fisher Scientific, 18080-044) and random hexamers (Thermo Fisher Scientific, N8080127), according to the producer’s protocol. All qPCR experiments were run on Roche LightCycler480 Real-Time PCR System using SYBR Green master mix (Roche, 04707516001, Basel, Switzerland). A mix of concentrated cDNA was used as a standard curve, two housekeeping genes (*Tbp* and *Sdha*) were used for normalization, and all samples were run in duplicates. The list of primer sequences can be found in Appendix A.

### 4.4. Western Blot

Cells were washed once with cold PBS on ice and scraped in Laemmli buffer. Cells were sonicated at 20% for 10 s to release the protein content using Branson Digital Sonifier Model S-450D (Marshall Scientific), centrifuged at maximum speed for 10 min at 4 °C, and stored at −80 °C until the gel could be run. The protein concentration was measured on NanoDrop (Thermo Fisher Scientific) A280, and 25–30 μg of total protein was loaded in each lane, diluted in Laemmli for the final volume of 30 μL with 2 μL 0.1 M DTT and 3 μL loading dye (0.01% (*w*/*v*) bromophenol blue). We ran the samples in NuPAGE Novex 4–12% Bis-Tris Protein Gels (Thermo Fisher Scientific, NP0321BOX) using 1× NuPAGE MES SDS running buffer (Thermo Fisher Scientific, NP0002) at 150 V for 70 min. We transferred the protein to Amersham Protran Nitrocellulose Blotting membrane 0.45 µm (GE Healthcare, 10600003, Chicago, IL, USA) in 1× NuPAGE transfer buffer (Thermo Fisher Scientific, NP0006) at 100 V for 1.5 h in a cold room. Membranes were shortly rinsed in PBS and blocked using 5% skim milk in PBS with 0.1% Tween-20 for 1 h at room temperature (RT) and next incubated with primary antibodies at dilution 1:1000 overnight at 4 °C. For blotting PGR, we used Progesterone Receptor A/B (D8Q2J) XP^®^ Rabbit mAb (Cell Signaling, 8757, Danvers, MA, USA). The following day, membranes were washed 3 × 10 min with 0.1% Tween-20 in PBS and incubated with HRP-conjugated secondary antibodies for 2 h at RT (donkey anti-Rabbit HRP (Thermo Fisher Scientific, A16023, 1:10,000)). Finally, the membranes were washed as previously, and the bands were visualized using Amersham ECL Prime Western Blotting Detection reagent (GE Healthcare, RPN2236) and ChemiDoc MP (Bio-Rad).

### 4.5. Immunofluorescence and Imaging

Cells were seeded on iBidi plates (ibidi, 8-well tissue culture treated #80826) at a density of 10–20 k cells/well, and fixed 24–48 h afterwards using 4% PFA for 15 min at RT. Next, we permeabilized the cells using 0.5% Triton in PBS for 10 min at RT and blocked the non-specific binding sites using 3% BSA in 0.1% Triton/PBS for 20 min at RT. We incubated cells with primary antibody solution (1% BSA in 0.1% Triton/PBS with antibodies) containing mouse anti-PGR (Santa Cruz Biotechnology, sc-810, Dallas, TX, USA) overnight at 4 °C. The following day, we washed cells 3 × 5 min using washing solution (1% BSA in 0.1% Triton/PBS) and incubated with fluorophore-conjugated secondary antibodies and nuclear dye DAPI in darkness for 2 h at RT. Finally, we washed the cells 3 × 5 min in washing solution and 1 × 5 min in PBS. The cells were either immediately imaged or stored at 4 °C and imaged within 1 month.

Mouse embryos were dissected and fixed in 4% PFA at RT for 30 min at RT. They were permeabilized using PBS/PVP + 1% (*v/v*) Triton X-100 for 10 min at RT, shaking. Embryos were blocked overnight at 4 °C in CAS-Block (Thermo Fisher Scientific, 008120). Embryos were incubated with the 1:100 diluted primary antibodies for 48 h at 4 °C, and the following antibodies were used: mouse anti-PGR (Santa Cruz Biotechnology, sc-810), goat anti-SOX2 (R&D Systems, AF2018, Minneapolis, MN, USA), and rabbit anti-BRACHYURY (R&D Systems, MAB20851). Embryos were washed 3 x 10 min with CAS-Block and then overnight at 4 °C. Next, they were incubated with secondary antibodies for 24 h in 4 °C. Embryos were washed 3 × 10 min in CAS-Block and then overnight at 4 °C. Afterwards, they were incubated with 1 μg/mL DAPI for 1 h at RT. Finally, embryos were washed 3 × 10 min at RT and dehydrated in ascending concentrations of methanol, eventually reaching 100%. Embryos were optically cleared in 1:2 solution of benzyl alcohol:benzyl benzoate (BABB). Cells and embryos were imaged using Leica TCS SP8 confocal microscope, and the images were analyzed using ImarisTM software (v9.3.0, Oxford Instruments, Abingdon, UK).

### 4.6. scRNA-Seq

Massively Parallel Single-Cell RNA-Sequencing technology (MARS-Seq) was performed for scRNA-Seq. EpiSCs were seeded at a density of 60 k cells per well of a six-well cell culture plate pre-coated with 15 µg/mL human Fibronectin and differentiated to PPS and ExM according to differentiation protocols described above. At 24 h and 48 h time-points, cells were dissociated using Accutase for 3 min at 37 °C and counted. A total of 0.5 × 10^6^ cells per condition were washed with ice-cold FACS buffer (10% (*v/v*) FBS in PBS) and resuspended in 1 mL ice-cold FACS buffer containing 1 µg/mL DAPI. Single cells were sorted into Eppendorf Polypropylene U-shaped 384-well Twin Tec PCR Microplates (Thermo Fisher Scientific, 10573035) containing 2 μL of lysis solution (0.2% (*v/v*) Triton X-100) supplemented with 0.4 U/μL RNasin Ribonuclease Inhibitor (Promega, N2515, Madison, WI, USA) and 400 nM indexed RT primer from group 1 (1–96 barcodes) or group 2 (97–192 barcodes). Wild-type (WT) EpiSCs were sorted into each plate, as spike-in control for batch-effect correction. Capture plates were prepared on the Bravo automated liquid handling robot station (Agilent, Santa Clara, CA, USA) using 384-filtered tips (Agilent, 19133-142). Index sorting was performed using a FACS Aria III cell sorter (BD Biosciences, San Jose, CA, USA) at the DanStem Flow Cytometry Platform (University of Copenhagen, Copenhagen, Denmark), sorting singlets and live cells only. Directly after sorting, plates were briefly centrifuged, snap-frozen on dry ice, and stored at −80 °C for further procedures. Semi-automated library preparation was carried out using 10–12 total cycles of PCR amplification and AMPure XP beads (Beckman Coulter Life Sciences, A63881, Indianapolis, IN, USA).

DNA concentration was measured with a Qubit Fluorometer (Thermo Fisher Scientific, Q32854) and fragment size was determined with a Fragment analyzer (Agilent). Libraries were paired-end sequenced on a Next-Seq 500 Sequencer (Illumina, San Diego, CA, USA) at the DanStem Genomics Platform (University of Copenhagen, Copenhagen, Denmark). Between 1146 and 1528 cells were sequenced per lane.

Data for PPS can be retrieved under the following accession number E-MTAB-12106, and the data for ExM under E-MTAB-12107.

### 4.7. Bulk RNA-Seq

EpiSCs were seeded at 60 k cells/well in six-well cell culture plates pre-coated with 15 µg/mL human Fibronectin and differentiated to PPS, APS, ExM, LM, and PM as described above. At chosen time points (either 24 h or 48 h), cells were lysed, and total RNA was extracted using the QIAGEN RNeasy Mini Kit according to the manufacturer’s instructions (Qiagen, 74106). RNA concentration was quantified by NanoDrop, and quality was verified with a Fragment analyzer. Library preparation was carried out using 0.5 μg of RNA with NEBNext Ultra II RNA library Prep Kit (New England BioLabs, E7770S, Ipswich, MA, USA) according to the manufacturer’s instructions. Libraries were amplified for 4 total PCR cycles and purified with AMPure XP beads. DNA concentration was measured with a Qubit Fluorometer, and fragment size was determined with a Fragment analyzer. All samples were sequenced in biological triplicates on a Next-Seq 500 Sequencer (Illumina) at the DanStem Genomics Platform (University of Copenhagen, Copenhagen, Denmark).

Bulk RNA-seq data can be accessed on ArrayExpress under E-MTAB-12104 accession number.

### 4.8. Cloning of the Overexpression Vectors and Lentiviral Packaging

RNA was extracted from WT EpiSCs using QIAGEN RNeasy Mini Kit (Qiagen, 74106), and 1 μg was reverse transcribed using SuperScript III enzyme (Thermo Fisher Scientific, 18080-044) and random hexamers (Thermo Fisher Scientific, N8080127) according to the manufacturer’s protocol. We PCR amplified *Pgr* sequences corresponding to isoforms A and B using the following primers:

PGR-B forward: 5′ CACC-ATGACTGAGCTGCAGGCAAAG 3′

PGR-B reverse: 5′ TCACTTTTTGTGAAAGAGGAGCGG 3′

PGR-A forward: 5′ CACC-ATGAGTCGGCCAGAGATCAAG 3′

PGR-A reverse: 5′ TCACTTTTTGTGAAAGAGGAGCG 3′.

We ran the PCR product on 1.5% agarose gel and extracted the DNA from gel using QIAquick Gel Extraction Kit (Qiagen, 28706). Purified PCR product was cloned into pENTR™/D-TOPO™ (Thermo Fisher Scientific, K240020) vector using TOP10™ chemically competent *E. coli*, which was part of the kit, according to the producer’s protocol. We then used Gateway™ LR Clonase™ Enzyme mix (Thermo Fisher Scientific, 11791019) and, following the manufacturer’s protocol, cloned the sequences encoding the two isoforms into destination vector pLEX307 (kindly provided by the Helin laboratory). The resulting constructs carrying the sequences encoding either PGR-A or PGR-B were verified by sequencing.

To prepare lentiviral vectors, HEK293T cells were seeded on 10 cm plastic cell culture dishes at a density of 6 × 10^6^ cells/dish. The following day cells were transfected using the following mix for each 10 cm dish: 15 μg pLEX307-PgrA or pLEX-PgrB, 10 μg packaging plasmid psPAX2, and 5 μg envelope-encoding pCMV-VSV-G with 90 μL 10 mg/mL PEI (1:3, DNA:PEI) in 500 μL Opti-MEM™ reduced serum medium (Thermo Fisher Scientific, 31985062). Transfection was performed in the late afternoon and medium was changed the following morning. Medium was collected 24 h and 48 h later for virus concentration. The virus was concentrated using Amicon^®^ Ultra 15 mL Centrifugal Filters (Merck Millipore, UFC900324). Briefly, collected medium was centrifuged for 15 min at maximum speed to remove cellular debris. Supernatant was loaded on the Amicon^®^ Ultra 15 mL Centrifugal Filters and centrifuged at maximum speed for 15–20 min. The loading of the media and centrifugation was repeated until all the virus-containing medium was passed through the filter and concentrated to 500–1000 μL total volume. The concentrated virus was stored in −80 °C for up to one year.

### 4.9. Isoform-Specific PGR Overexpression during Differentiation

Cells were seeded on fibronectin-coated 6-well plates at a density of 100 k cells/well in the appropriate culture media. The following day cells were transduced at MOI > 0.7, and the following day medium was changed to EpiSC medium containing 1 μg/mL puromycin (Sigma Aldrich, P8833, Saint Louis, MO, USA). The selection pressure was maintained for the following two days, when the cells were passaged and seeded for differentiation experiment on 6-well plates at a density 100 k cells/well. The following day medium was changed to differentiation medium. Cell lysates were collected 24 h and 48 h after induction of mesoderm differentiation for the analysis.

### 4.10. Cell Lines Genotyping and Karyotyping

Cell lines were genotyped using GoTaq^®^ Green Master Mix (Promega, M7122). At least 10 ng of gDNA was used for each PCR reaction. The product of the reaction was run on 2% agarose gel, and the resulting bands were imaged using ChemiDoc MP (Bio-Rad). For karyotyping cells were sent to Cell Guidance Systems, Cambridge, UK.

### 4.11. Mice

All animals used in this study were maintained in laboratory animal housing facilities at macroenvironmental temperature and humidity ranges of 20–24 °C and 45–65%, respectively, with a 12/12 h light/dark cycle. All animal work was carried out in accordance with European legislation. All work was authorized by and carried out under the Project License 2017-15-0201-01255 issued by the Danish Regulatory Authority. Mouse embryos were dissected at stages E6.5-7.5, with the stage described as E6.75 dissected in the evening.

## Figures and Tables

**Figure 3 ijms-23-10307-f003:**
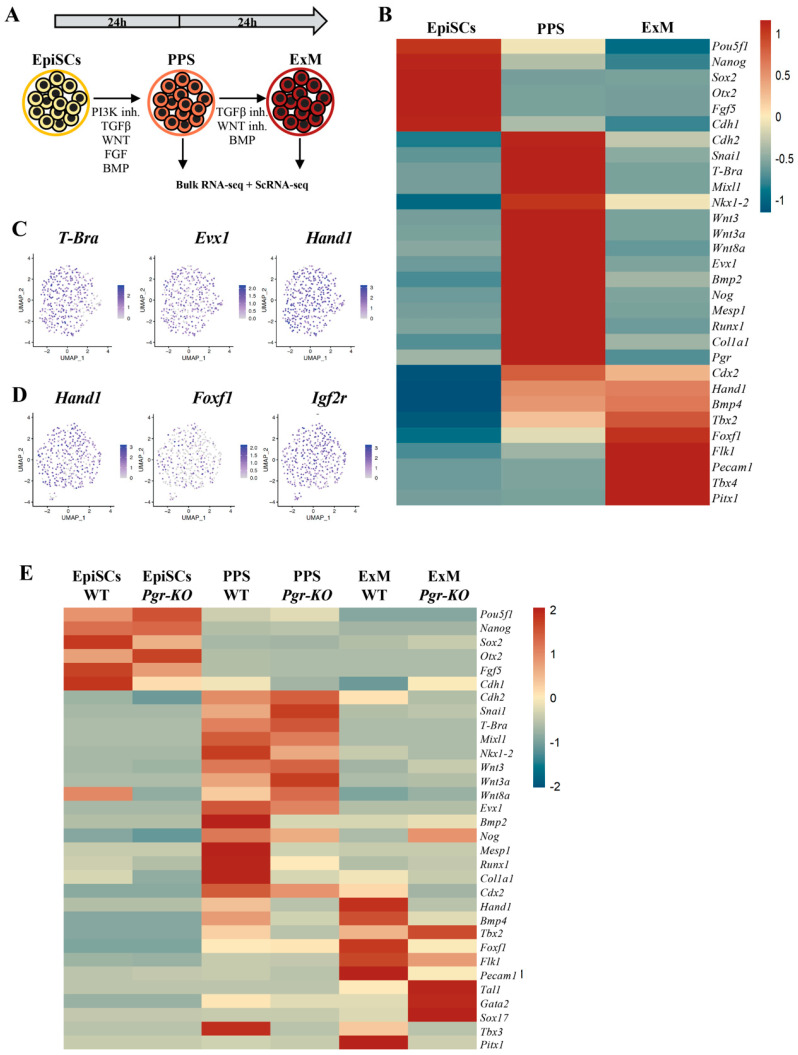
*Pgr* modulates PPS induction and extraembryonic mesoderm (ExM) formation. (**A**) Schematic illustration of the extraembryonic differentiation protocol. EpiSCs are initially subjected to PPS induction, and thereafter exposed to BMP for additional 24 h to induce ExM differentiation. High levels of BMP restrict PS cells towards extraembryonic mesoderm-like fate. (**B**) Heatmap displaying relative expression of selected differentially regulated genes in EpiSCs, PPS, and ExM (RNA-Seq, *p* ≤ 0.05, fold-change ≥ 1.5). (**C**) UMAP plots showing gene expression distribution in PPS cells. Individual cells are colored by expression of key posterior primitive streak markers (*T-Bra*, *Exv1*, and *Hand1*). (**D**) UMAP plots showing gene expression distribution in ExM. Individual cells are colored by the expression of key extraembryonic mesoderm marker genes (*Hand1*, *Foxf1*, and *Igrf2*). (**E**) Differential expression analysis of WT and *Pgr-KO* EpiSCs differentiated towards PPS and ExM revealing downregulation of PPS markers (*Hand1*, *Col1a1*, and *Bmp2*) and a stronger effect in ExM (*Hand1*, *Foxf1*, *Pecam1*, *Flk1*, and *Bmp4)*. Upregulation of *Tal1*, *Gata2*, and *Sox17* in *Pgr-KO* indicates a shift of ExM towards a hemangioblast fate (RNA-Seq, *p* ≤ 0.05, fold-change ≥ 1.5).

**Figure 4 ijms-23-10307-f004:**
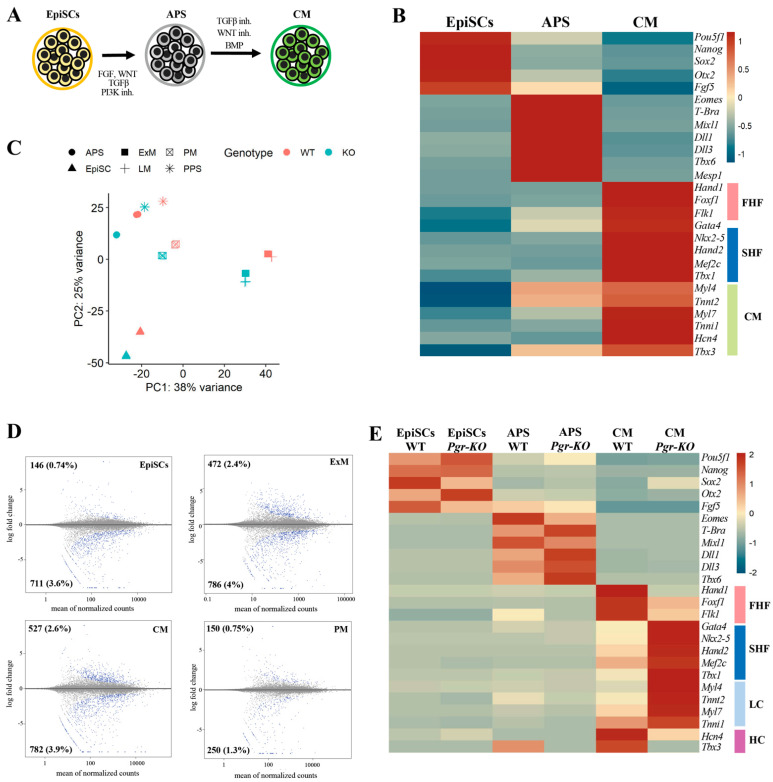
PGR differentially regulates FHF and SHF cardiac progenitors. (**A**) Schematic illustration of the cardiac mesoderm (CM) differentiation protocol. EpiSCs are initially subjected to APS induction, and thereafter exposed to BMP for 24 h to induce CM differentiation. (**B**) Heatmap displaying relative expression of selected differentially regulated genes in EpiSCs, APS, and CM (RNA-Seq, *p* ≤ 0.05, fold-change ≥ 1.5). (**C**) PCA plots clustering WT and *Pgr-KO* cells subjected to differentiation conditions: EpiSCs, PPS, APS, extraembryonic mesoderm (ExM), cardiac mesoderm (CM), and paraxial mesoderm (PM). CM and ExM are affected to a higher extent by *Pgr* depletion. (**D**) MA plots of WT and *Pgr-KO* comparisons in EpiSCs, ExM, CM, and PM showing the highest number of differentially expressed genes in CM and ExM. Blue dots indicate differentially expressed genes with *p* ≤ 0.05, fold-change ≥ 1.5 (**E**) Differential expression analysis of WT and *Pgr-KO* EpiSCs differentiated towards APS and CM, showing modest effects of loss of *Pgr* on APS induction, but stronger consequences within CM differentiation. In CM, the FHF cardiac markers (*Hand1*, *Flk1*, and *Foxf1*) are downregulated, while the SHF markers (*Gata4*, *Nkx2-5*, *Hand2*, *Mef2*, and *Tbx1*) are upregulated, revealing a propension of *Pgr-KO* cells to acquire a cardio-pharyngeal fate. First heart field (FHF, ligh pink), second heart field (SHF, blue), late cardiomyocytes (LC, light blue), and heart conductive (HC) makers (pink).

**Figure 5 ijms-23-10307-f005:**
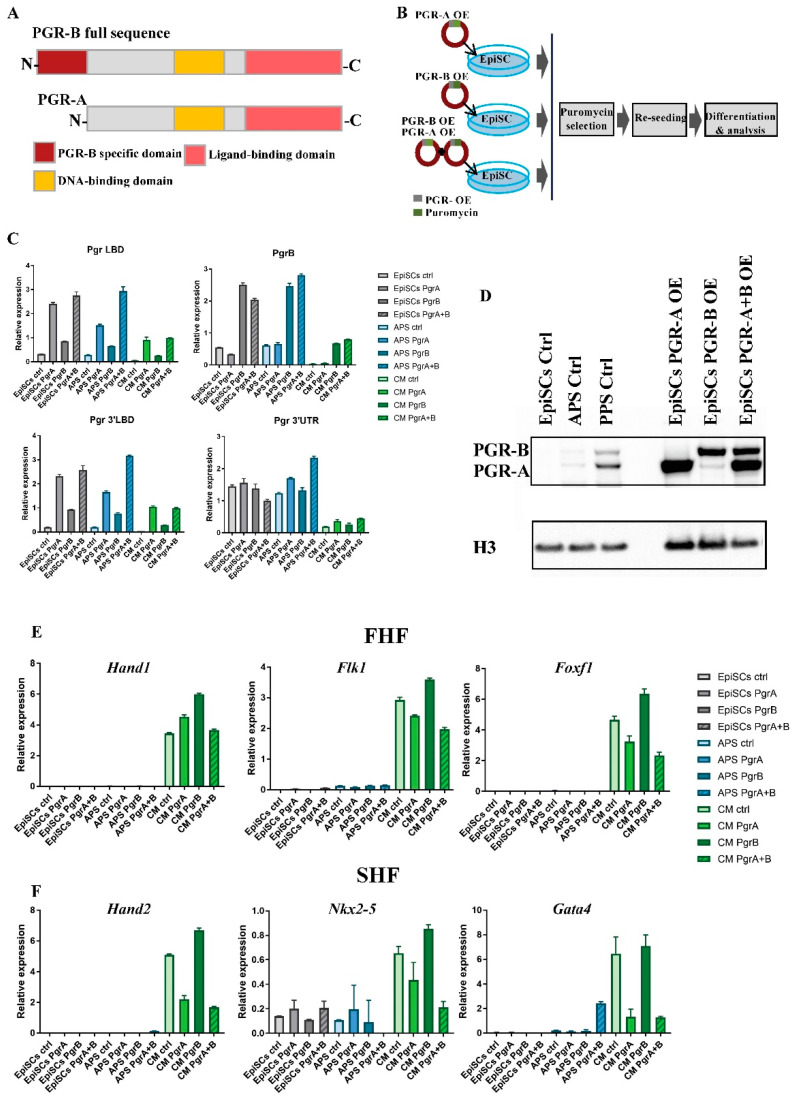
PRG-B and PGR-A selectively regulate FHF vs. SHF cardiac differentiation. (**A**) Illustration depicting PGR protein structure. The long form of PGR, PGR-B, includes the PGR-B-specific domain (dark red), the DNA-binding domain (yellow), and the ligand-binding domain (light red). The short form of PGR lacks the N-terminal PRG-B domain. (**B**) Schematic of the overexpression assay. The sequences of both full-length PGR-B and shorter PGR-A were cloned into a lentiviral vector under the *EF1α* promoter. After transduction, cells undergo puromycin selection and differentiation. (**C**) Relative mRNA expression of *Pgr* isoforms measured by RT-qPCR along differentiation of EpiSCs to APS and CM, confirming overexpression of transduced PGR. (**D**) Western blot showing the levels of overexpressed (OE) PGR isoforms in EpiSCs. Overexpression of specific isoforms is higher than the endogenous levels in EpiSC, APS, and PPS. (**E**) RT-qPCR analyses of cells overexpressing PGR isoforms showing mild upregulation of the FHF signature *Hand1*, *Flk1*, and *Foxf1* due to PGR-B overexpression. (**F**) RT-qPCR analyses indicate decreased levels for the SHF signature markers (*Hand2*, *Nkx2-5*, and *Gata4*) by overexpressing PGR-A isoform.

**Figure 6 ijms-23-10307-f006:**
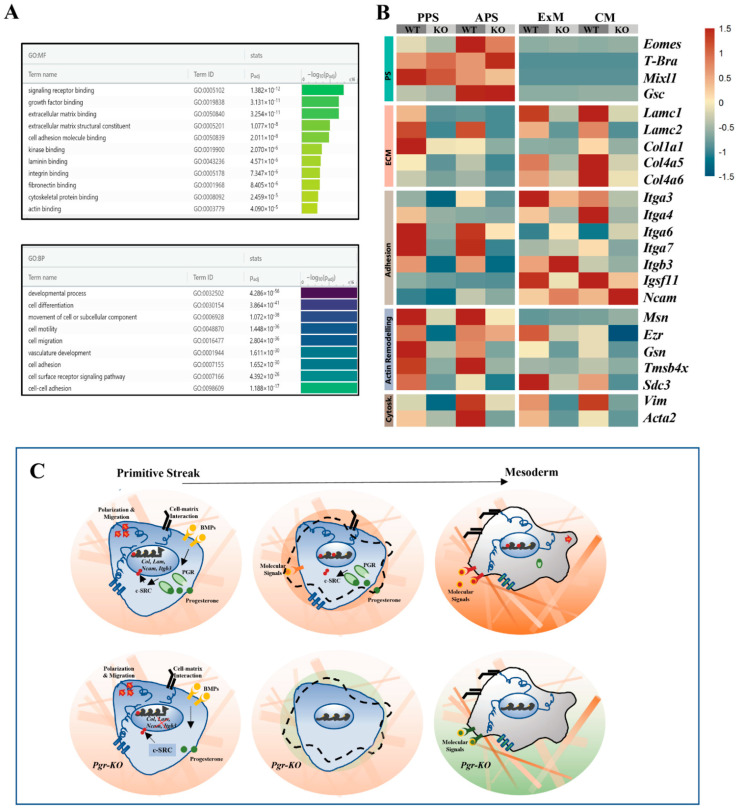
PGR influences cell-ECM, cell-cell interactions and cytoskeletal remodeling of mesoderm progenitors. (**A**) GO-term enrichment analysis of genes differentially expressed in WT and *Pgr-KO* PPS displays significant overrepresentation of terms associated with ECM interactions, cell adhesion, and cytoskeletal protein binding (Fisher’s exact test with Bonferroni correction). (**B**) Heatmap indicating downregulation of Laminins, Collagens, and actin remodeling transcripts in APS and PPS with moderate changes in the PS signature genes. In ExM and CM, adhesion and ECM genes are downregulated in *Pgr-KO*, with the exception of *Itgb3*, which is specifically upregulated in the ExM and *Ncam*1, which is upregulated in CM. (**C**) *PGR-KO* cells show altered expression of genes related to cellular adhesion, cytoskeleton remodeling, and ECM composition. Changes in the cellular interactions with the direct environment alter the differentiation program and affect the balanced production of mesoderm types.

**Figure 7 ijms-23-10307-f007:**
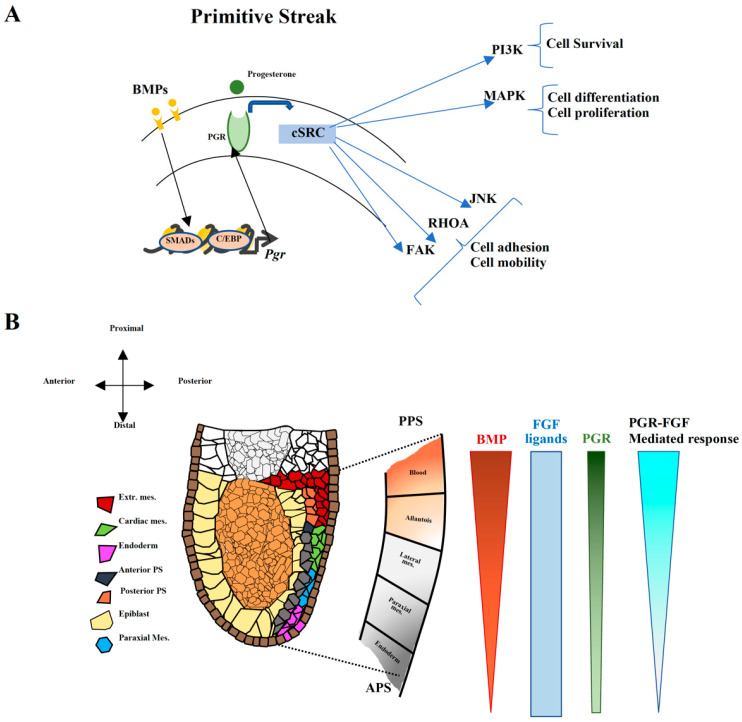
BMP-induced PGR expression fine-tunes and amplifies FGF response during cell specification. (**A**) Illustration showing a proposed molecular mechanism of PGR function during differentiation. As described previously, BMP induces PGR upregulation [50]. PGR, in turn, interacts with c-SRC [24] and activates its downstream targets: PI3K, MAPK, JNK, RHOA and FAK, thus influencing cell survival, differentiation, proliferation, adhesion, and motility. (**B**) Illustration depicting ligand gradients described in the gastrulating mouse embryo at around E6.5 with BMP expressed in posterior–anterior gradient and modulating the expression of PGR along the same axis. We propose that this PGR gradient establishes a gradient for FGF response in the early mouse embryo, as shown in A, thus influencing cell fate decisions during mesoderm development, specifically ExM and CM specification.

## Data Availability

The data is available under the following accession numbers on ArrayExpress: E-MTAB-12104 (RNAseq data for WT and *Pgr-KO* cells), E-MTAB-12106 (scRNA-seq for PPS), E-MTAB-12107 (scRNA-seq for ExM).

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
