# Peer review of "Progesterone Receptor Modulates Extraembryonic Mesoderm and Cardiac Progenitor Specification during Mouse Gastrulation"

_ijms, 2022, doi:10.3390/ijms231810307_

Round 1

Reviewer 1 Report

This manuscript provides important information for the scientific community.

Author Response

We thank the referee for their positive assessment of our paper and the comments and suggestions that have helped us improve our manuscript.

This manuscript provides important information for the scientific community.

Indeed, the progesterone signalling activity is of particular importance due to its role in stem cell differentiation, cellular homeostasis, and cancer. Thus, our findings have broad relevance for disease modelling, regenerative medicine, and cancer biology, in addition to being of specific interest to the stem cell biology field.

In light of these comments, we checked for English misspellings and added three new references to the manuscript. We realized that in the discussion, we mentioned possible stem cell-based therapies without providing proper references.

Thus we added the following references:

  1. Kim SW, Woo HJ, Kim EH, Kim HS, Suh HN, Kim SH, Song JJ, Wulansari N, Kang M, Choi SY et al: Neural stem cells derived from human midbrain organoids as a stable source for treating Parkinson's disease: Midbrain organoid-NSCs (Og-NSC) as a stable source for PD treatment. Prog Neurobiol 2021, 204:102086.10.1016/j.pneurobio.2021.102086
  2. Siehler J, Blochinger AK, Meier M, Lickert H: Engineering islets from stem cells for advanced therapies of diabetes. Nat Rev Drug Discov 2021, 20(12):920-940.10.1038/s41573-021-00262-w
  3. Ai X, Yan B, Witman N, Gong Y, Yang L, Tan Y, Chen Y, Liu M, Lu T, Luo R et al: Transient secretion of VEGF protein from transplanted hiPSC-CMs enhances engraftment and improves rat heart function post MI. Mol Ther 2022.10.1016/j.ymthe.2022.08.012

The revised manuscript is attached.

Reviewer 2 Report

The manuscript titled Progesterone receptor modulates extraembryonic mesoderm and cardiac progenitor specification during mouse gastrulation by Drozd et al. invesitgates in vitro differentiation of embryonic stem cells to determine the role of the progesterone receptor. The results are cumbersome and difficult to interpret. Much of the results includes discussion and implications of the result. While this style reads well, it becomes confusing because it is difficult to decipher what is in the present study and what was done previously. I feel for the reader, it would be best to rewrite the results with clear and concise results of the present study and restrict interpretation to the discussion section.

Further it is not clear how embryos were dissected and used in the present study.

Author Response

We thank the reviewer for their comments and acknowledge their criticisms. We have rephrased the result chapter to make it more straightforward, as suggested by the referee.

Much of the results includes discussion and implications of the result. While this style reads well, it becomes confusing because it is difficult to decipher what is in the present study and what was done previously. I feel for the reader, it would be best to rewrite the results with clear and concise results of the present study and restrict interpretation to the discussion section.

We have eliminated the unnecessary comments in the results chapter. However, we keep some interpretations to maintain the flow and provide a rationale for the experiments performed. We write the result in the present tense so that it will be more evident when in the text we describe our experiments or if we refer to published work. We removed panel C from Figure 6, illustrating a possible molecular mechanism. We discussed the potential model in the discussion.

We thank the reviewer for raising this important question making it apparent that our original manuscript was unclear.

Further it is not clear how embryos were dissected and used in the present study.

We added a paragraph describing the mouse and the embryo dissection in the Material and Methods:

4.11. Mice.

All animals used in this study were maintained in laboratory animal housing facilities at macroenvironmental temperature and humidity ranges of 20–24°C and 45–65%, respectively, with a 12/12 h light/dark cycle. All animal work was carried out in accordance with European legislation. All work was authorized by and carried out under the Project License 2017-15-0201-01255 issued by the Danish Regulatory Authority. Mouse embryos were dissected at stages E6.5-7.5, with the stage described as E6.75 dissected in the evening.

Further dissection description is included in Methods section 4.5. Immunofluorescence and imaging.

The attached revised manuscript contains all the edited parts and is followed by track changes.
